# Biodegradable Polymers as Drug Delivery Systems for Bone Regeneration

**DOI:** 10.3390/pharmaceutics12020095

**Published:** 2020-01-24

**Authors:** Kaoru Aoki, Naoto Saito

**Affiliations:** 1Physical Therapy Division, School of Health Sciences, Shinshu University, Asahi 3-1-1, Matsumoto, Nagano 390-8621, Japan; kin29men@shinshu-u.ac.jp; 2Institute for Biomedical Sciences, Interdisciplinary Cluster for Cutting Edge Research, Shinshu University, Asahi 3-1-1, Matsumoto, Nagano 390-8621, Japan

**Keywords:** biodegradable polymer, drug delivery system, bone regeneration

## Abstract

Regenerative medicine has been widely researched for the treatment of bone defects. In the field of bone regenerative medicine, signaling molecules and the use of scaffolds are of particular importance as drug delivery systems (DDS) or carriers for cell differentiation, and various materials have been explored for their potential use. Although calcium phosphates such as hydroxyapatite and tricalcium phosphate are clinically used as synthetic scaffold material for bone regeneration, biodegradable materials have attracted much attention in recent years for their clinical application as scaffolds due their ability to facilitate rapid localized absorption and replacement with autologous bone. In this review, we introduce the types, features, and performance characteristics of biodegradable polymer scaffolds in their role as DDS for bone regeneration therapy.

## 1. Introduction

Pluripotent cells—such as induced pluripotent stem cells (iPSCs) and embryonic stem cells (ESCs)—for the regeneration of lost tissues and organs hold great promise in the field of regenerative medicine [1,2]. Tissue regeneration is difficult to achieve with cells alone and requires a combination of cells, scaffolds, and signaling molecules to play their respective roles [3].

Severe trauma and malignant bone tumors can cause large bone defects, wherein a substantial part of the bone is removed [4,5,6]. A variety of techniques are available for treating bone defects, including autologous bone grafting [7,8], allogenic bone grafting [9,10,11], synthetic bone grafting [12,13,14,15,16], artificial joint replacement [17,18], and the induced membrane technique [19]. The method of treatment is chosen according to several factors that include the size and position of the defect [4].

Large bone defects which are too large to repair with bone tissue may require cobalt–chromium or titanium alloy prostheses. Unlike conventional artificial joint replacement surgery for elderly patients [20,21], metallic megaprostheses have been used for the reconstruction of bone defects from tumor resection or severe trauma. The use of megaprostheses has a higher complication rate than conventional arthroplasty and may require revision surgery due to infection, loosening, and wear over time [22]. Artificial joint replacement surgery is often difficult to perform due to scar and joint contracture after the initial operation, leading to longer operation time, increase in blood loss, and greater invasiveness. Revision surgery may further increase the size of bone and soft tissue defects, resulting in a narrower range of motion, reduced walking ability, and decreased activities of daily living (ADL).

Bone defects are also treated with autologous or allogenic bone grafts. Because healthy tissues are damaged when autologous bone is harvested from fibular or iliac donor sites, the additional procedure may cause postoperative pain at the donor site, difficulties in walking, and problems related to limited graft availability [23,24,25,26]. On the other hand, allogenic bone from cadaveric or living donors can be procured in sufficient quantities where infrastructure and resources are available for bone banks [27,28,29,30,31]. However, allogenic bone grafts are more prone to immune rejection and sometimes result in nonunion, resorption, revision surgery, or secondary fracture [32,33,34].

The induced membrane technique involves the placement of a cement spacer into the bone defect to induce the formation of a biological membrane around the cement. The cement spacer is removed after six to eight weeks, and a combination of granular bone substitute and autologous bone are placed into the cavity of the induced membrane to promote the regeneration of bone. Due to the periosteum-like membrane surrounding the defect, relatively large bone defects can be treated with less autologous bone compared to conventional autologous bone grafts. However, the induced membrane technique requires a two-stage procedure, and the amount of bone defect that can be regenerated remains limited [35].

As described above, there are various methods for treating bone defects, each with its own advantages and disadvantages. The need for safe and reliable methods of bone regeneration remains a focus in regenerative medicine research. The use of synthetic bone substitutes does not require allogenic or autogenous bone grafts to repair bone defects, and sufficient material can be obtained that functions as a scaffold for the regeneration of bone. A variety of scaffolds for bone tissue regeneration are being researched and developed, including materials made of tricalcium phosphate (TCP) and hydroxyapatite (HA), which are similar to the inorganic component found in the bone matrix [36,37], as well as biodegradable polymers that can be absorbed and replaced by new bone (Table 1).

When a biodegradable polymer is implanted into a bone defect as a scaffold enhanced with signaling molecules to promote bone formation, the cells differentiate into bone and are induced and expanded by the signaling molecules released from the polymer. As the autologous bone is regenerated by these cells, the polymer is degraded and replaced by new bone to repair the bone defect (Figure 1).

In this review, we will introduce biodegradable polymers that are used as scaffolding materials in regenerative medicine of the bone, in addition to drug delivery systems for signaling molecules that can induce bone regeneration.

## 2. The Use of Biodegradable Polymers as Scaffolds

### 2.1. Bio-Based Polymers

There has been considerable research into biologically derived materials that exhibit excellent biocompatibility and biological safety [38]. In particular, collagen is used as a biomaterial in many clinical applications, such as artificial skin [39], hemostatic agents [40], contact lenses [41], and vascular prosthesis [42].

Collagen is widely observed in vivo as a constituent of fibrous tissue, cartilage, and bone. In bone, more than 90% of the protein in the bone matrix is composed of collagen. Type I collagen, the main collagen present in bone, is formed by osteoblasts. The N- and C-termini of procollagen, which are precursors of collagen formed in the cell, are secreted outside of the cell and subsequently cleaved by protease to become a collagen molecule. The collagen molecules self-associate into aggregates to become collagen fibrils, and cross-linking bonds are formed between collagen molecules in collagen fibrils, resulting in cross-linked collagen fibrils [43] (Figure 2). Calcium phosphate is deposited on collagen fibers that are formed by osteoblasts during the development, regeneration, and remodeling of bone, thereby completing the process of creating a strong bone matrix [44]. Because collagen is an important component of hard tissues and fibrous tissues of in vivo models, there has been a substantial amount of research in regenerative medicine into its use as a scaffold and its potential role in the regeneration of the myocardium [45], bladder [46], and ligaments [47]. With regard to the regeneration of bone tissue, much research has been actively conducted on animal models, and good bone formation has been obtained at an experimental level (Figure 3). Carstens et al. [48] used collagen as a scaffold to regenerate non-weight-bearing bones, such as porcine maxilla and mandible models. Venugopal et al. [49] produced a composite of type I collagen and HA. Human fetal osteoblast cells proliferated well on this scaffold, and an evaluation of mineralization with Alizarin Red S staining showed better mineralization than collagen nanofiber scaffolds without HA [50]. Yeo et al. [51] produced a porous three-dimensional composite of β-tricalcium phosphate (βTCP) and polycaprolactone (PCL) which are components of the bone matrix, and embedded collagen nanofibers in the pores of the scaffold. In this scaffold, MTT (3-(4,5-dimethylthiazol-2-yl)-2,5-diphenyl tetrazolium bromide) assay demonstrated good cell proliferation. The assay was used to evaluate the cell proliferation of human MG-63 osteoblast-like cells as opposed to βTCP/PCL scaffold without embedded collagen nanofibers [52].

Sponge-like HA/type I collagen composite is commercially available and clinically used as artificial bone. It can handle all types of bone defects and can be quickly replaced by autologous bone [16]. However, its initial strength as artificial bone is weak, and inflammatory reactions such as swelling, exudation, and redness of a surgical wound may occur if the bone is in near proximity to the surface layer such as a finger bone, and its area of use is limited.

Collagen fibers have a triple-helix structure consisting of polypeptide chains and gelatin in its modified and degraded form. Gelatin is widely used in food and cosmetics, and in the medical field, gelatin sponge is used as a hemostatic agent for surgery and trauma care [53] and an embolic material for endovascular use [54,55]. Gelatin also features excellent biodegradability and is studied for its use as a scaffold for regenerative medicine. Yokota et al. [56] incorporated recombinant human bone morphogenetic protein (rhBMP-2) into a gelatin sponge coated with polylactic-*co*-glycolic acid (PLGA) and observed ectopic bone formation after its dorsal subcutaneous implantation in Fischer rats. Hokugo et al. [57] used gelatin hydrogel as a carrier for platelet-rich plasma (PRP) to repair a rabbit critical-size defect in the ulna. Rohanizadeh et al. [58] examined the effectiveness of gelatin sponge, a commercially available hemostatic agent, as a scaffold. The number of cells, the activity of alkaline phosphatase (ALP) as a marker of bone formation, and the entry of cells into the pore of the sponge were observed upon culturing MG-63 cells onto a gelatin sponge. The results demonstrated the ability of gelatin sponge to support cell growth and suggested that it may be useful as a scaffold for bone regeneration.

Although collagen is one of the most commonly used naturally-derived scaffold materials, other polymers such as cellulose, chitosan, and hyaluronic acid are also being considered as potential scaffolds. Cellulose is a main component of plant fibers and is known as a raw material for paper; however, it is a polysaccharide formed by glycosidic bonds (Figure 4). Cellulose is also a material with excellent biocompatibility [59], and the cellulose/hyaluronic acid complex has been clinically applied as a barrier material for preventing postoperative adhesion [60]. As a scaffold for regenerative medicine, research has been conducted on the use of methyl cellulose for cartilage regeneration [61]. Chakraborty et al. [62] evaluated web-shaped cellulose nanofibers as a scaffold for bone regeneration and reported good results on MTT assay and scanning electron microscopy (SEM) when the MC3T3-E1 osteoblast cell line was cultured on the scaffold.

Chitosan is a structural element obtained by the deacetylation of chitin, which is a polysaccharide that can be found in the exoskeleton of crustaceans such as crabs and shrimp [63] (Figure 4). By immersing and heating chitin in an aqueous sodium hydroxide solution with a concentration above 40%, chitin can be deacetylated by 75% or more. Increasing the temperature and alkali concentration can increase the degree of deacetylation of chitosan; however, full deacetylation cannot be achieved by this method alone. To obtain 100% deacetylated chitosan, the partially deacetylated chitosan is filtered from the aqueous sodium hydroxide solution, thoroughly rinsed with water to remove the sodium hydroxide, and subsequently dried. Chitosan-derived biomaterials have been studied as scaffolds for nerve regeneration [64] and skin regeneration [65]. Sharifi et al. [66] synthesized a scaffold consisting of nanofibers of polycaprolactone (PCL)/chitosan composite, and subsequently performed an MTT assay with MG63 cells to evaluate its cell proliferation. Liu et al. [67] produced a scaffold made of chitosan nanofiber combined with HA nanoparticles. Bone marrow-derived mesenchymal stem cell (BMSC) were cultured onto the scaffold and evaluated by ALP staining. The HA/chitosan nanofiber scaffolds showed better staining properties than those of HA/chitosan membranous scaffolds. In addition, repair of bone defects was confirmed by implanting the HA/chitosan nanofiber scaffolds into critical bone defects of rats.

Hyaluronic acid is a type of linear glucosaminoglycan (Figure 4), and has a high water retention property as a component of cartilage and joint fluid. The acid can be obtained by extraction from cockscombs of chickens or produced from lactic acid bacteria, and it is used for the treatment of osteoarthritis and rheumatoid arthritis due to its viscosity and lubricating properties [68]. Hyaluronic acid is also used in cosmetics as a moisturizer due to its high water retention [69]. As a drug delivery system (DDS), Yan et al. [70] prepared a hyaluronic acid hydrogel from hyaluronic acid hydrazide derivatives. They combined rhBMP-2 to form ectopic bone beneath the dorsal muscle fascia of rats. The study successfully made neutral hydrogels at pH 7 and acidic hydrogels at pH 4.5, and their osteogenic potential were compared. In these experiments, acidic hydrogels showed a higher volume of bone and better bone formation, and the researchers stated that the release of rhBMP-2 from hyaluronic acid hydrogel is regulated by electrostatic and van der Waals forces and can be adjusted by manipulating pH levels. In addition, Paidikondala et al. prepared a hydrazone-cross-linked hyaluronic acid-based hydrogel comprising of hydrazone and cross-linked hyaluronic acid, which was combined with rhBMP-2 and injected into the dorsal muscle fascia of rats, thereby inducing the formation of ectopic bone [71]. This suggested the possibility of bone regeneration therapy without the need for surgical intervention, as bone formation scaffolds can be administered by injection.

Natural polymers are used in foods and cosmetics, which may suggest their excellent biocompatibility when used as biomaterials. There are many studies that report on their useful properties such as biocompatibility and degradability, and they may be suitable as a scaffold material for bone regenerative medicine. However, there is a possibility of an immune reaction due to disease transfer or xenogenicity [72,73]. Considering that there are reports of allergic reactions to injections and food products, inflammatory reactions, and pulmonary complications, the material should be used with some degree of caution [74,75,76]. A summary of the literature discussed in this chapter is shown in Table 2.

### 2.2. Synthetic Polymers

Langer, Vacanti, and colleagues seeded chondrocytes that were isolated from bovine articular cartilage onto a synthetic polymer scaffold made of polyglycolic acid–polylactic acid in the shape of a human ear, which was subcutaneously implanted and grown in the dorsal region of an immunodeficient nude mouse [77,78]. Through their work, scaffolds were popularized in the field of regenerative medicine.

Because biodegradable synthetic polymers can degrade through hydrolysis and get absorbed in vivo, synthetic polymers are gaining popularity as scaffold material for bone regeneration. We have previously developed and reported on a biodegradable polymer that combines poly lactic acid-*p*-dioxanone-polyethylene glycol block copolymer (PLA-DX-PEG) and rhBMP-2, and observed its biodegradable properties in vitro (Figure 5). The PLA-DX-PEG implant was combined with rhBMP-2 to produce a composite implant, and we observed ectopic bone formation in the dorsal fascia of a mouse model [79]. In addition, we produced a critical-size bone defect in the ulna of a rabbit and transplanted a PLA-DX-PEG implant containing rhBMP-2. The implants were implanted in 1-cm-distance rabbit ulnar bone defects, and these defects were examined after 2, 3, 4, 6, 9, and 13 weeks using radiographical methods. In the control group with defects that were not filled with the implant, the defects were not repaired at three months postoperatively (Figure 6a). The bone defects were also not repaired in the group with PLA-DX-PEG implants alone (Figure 6b) and the group with PLA-DX-PEG implants combined with a small quantity of rhBMP-2 (Figure 6c). In the group with PLA-DX-PEG implants with a large quantity of rhBMP-2, the bone defect was repaired at two months (Figure 6d). Because PLA-DX-PEG implants alone have low osteoconductivity, signaling molecules such as rhBMP-2 are required to repair large bone defects. Using a PLA-DX-PEG implant as a scaffold, we observed that the repair of the bone defect and its strength are dependent on the dose of rhBMP-2.

Other synthetic polymers such as PCL, PLGA, and poly (vinyl alcohol) (PVA) have been studied for their functional capacity as scaffolds. PCL is widely used as a thermoplastic with a low melting point for industrial applications. PCL is a polymer of ε-polycaprolactone [80] (Figure 7) and has excellent biocompatibility [81]. Wang et al. [82] added nanosilicate to PCL to synthesize a scaffold consisting of nanofibers with a diameter of several hundred nanometers. In the MC3T3-E1 osteoblast cell line culture, the cell viability and ALP activity increased due to the dose dependency of the nanosilicate added to the PCL. MC3T3-E1 cells were cultured onto a nanosilicate/PCL scaffold and transplanted subcutaneously on the dorsal area of a rat. At four weeks postoperatively, the nanosilicate/PCL composite scaffold exhibited more osteogenic activity than the scaffold with PCL alone. The expression of osteocalcin (OCN), a biochemical marker for bone formation, was strong [83].

PLGA is also a synthetic polymer with excellent biocompatibility, and there has been continued research on its DDS and use as scaffolds. PLGA is a copolymer of lactic acid and glycolic acid (Figure 7). Yang et al. [84] reported on a scaffold composed of nanofibers with a diameter ranging from 500 to 800 nm by adding nanosilicate to PLGA. Osteoblast-like cells (SaOS-2 cells) were cultured onto this scaffold and ALP activity was evaluated using Alizarin Red S staining. The authors demonstrated that nanosilicate/PLGA scaffolds promoted more osteogenic differentiation than scaffolds with PLGA alone.

The PLA-DX-PEG we evaluated is a copolymer of polylactic acid (PLA) and polyethylene glycol (PEG), and PLA is a biodegradable polymer used in food trays that are allowed to come into contact with food and agricultural films. Lactic acid consists of two optical isomers, l-lactic acid and d-lactic acid, and the polymer consisting of l-lactic acid alone is called poly-l-lactic acid (PLLA). Zhang et al. [85] fabricated a nanofibrous layer of poly-l-lactic acid (PLLA) with a layer of collagen to synthesize a bi-layer collagen/PLLA scaffold. From the BMSC cultured onto this scaffold, a stronger OCN gene expression was observed compared to the BMSC that was cultured onto the collagen scaffold. Moreover, in an experiment in which a collagen/PLLA scaffold was implanted in a bone and cartilage defect on the articular surface of the distal femoral of a rabbit, good regeneration of the cartilaginous bone was found in an evaluation using the International Cartilage Repair Society Visual Histological Assessment Scale compared to a group implanted with a collagen-only scaffold [86].

PVA is highly hydrophilic, easily dissolved in vivo, and has a structure in which vinyl alcohol is polymerized (Figure 7) [87]. Hydrogel-like PVA is suitable for making composites with other materials, and Enayati et al. [88] developed a PVA/HA scaffold in which HA nanoparticles are combined with PVA. MG63 cells were cultured on PVA/HA scaffolds and their effects were evaluated for bone formation. The MTT assay demonstrated that the cell viability was not significantly different from that of the PVA scaffold without HA; however, with the Alizarin Red S staining and ALP activity, the PVA/HA scaffold provided better results and promoted osteoblast differentiation. Although PVA is a material with excellent biocompatibility, PVA itself is bio-inert [89], and it has been suggested that the differentiation of osteoblasts was promoted by the effect of HA.

Both bio-derived and synthetic polymers are often used as a single type of material or in a combination of multiple materials. Various combinations of scaffolds such as composites of biodegradable polymers and composites of biodegradable polymers with inorganic materials have been studied. Zhang et al. [85] used BMSCs that were cultured onto collagen/PLLA composite scaffolds and reported better expression of osteogenic markers than with collagen-only scaffolds. Enayati et al. [88] demonstrated that PVA/HA composite scaffolds performed better in terms of Alvarin Red S staining and ALP activity compared to PVA-only scaffolds. The type of polymer, the method of creating the composite, and the structure may be related to the ability to form bones in polymer composites; however, there are innumerable combinations of materials, and the optimal composite for this application remains unknown. We believe that there is room to improve the performance of polymer composites as scaffolding material by taking advantage of the characteristics and advantages of each material.

A summary of the literature discussed in this section is shown in Table 3.

## 3. Biodegradable Scaffolds as Drug Delivery Systems

Although calcium phosphate-based artificial bones (TCP, HA) that are currently used for treating bone defects have osteoconductive properties and are relatively effective in repairing small defects on their own, there is still a limit to the size of the bone defect it is able to treat [15]. In contrast to calcium phosphate scaffolds that are osteoconductive, both natural and synthetic polymers are generally not osteoconductive; therefore, adding signaling molecules and osteoconductive cells to these scaffolds may enable these materials to function as a drug delivery system (DDS) that could improve the efficiency of bone regeneration.

BMP-2 is the most common among the signaling molecules used for bone regeneration. BMP-2 is a powerful osteoinductive factor and has been clinically applied to treat fractures and bone defects [90,91]. In order to make BMP-2 function efficiently in the affected area, the performance of the scaffold is considered important. Deng et al. [92] created 3D printing scaffolds by adding chitosan-coated rhBMP-2 to PLGA/HA and evaluated their release rate in addition to their osteogenic potential. In the in vitro experiment, the PLGA/HA/chitosan/rhBMP-2 scaffold in the culture medium degraded over a two-week period, and the gradual release of rhBMP-2 from the scaffold was observed over the course of one month. Furthermore, in the in vivo experiment, the scaffold was able to successfully repair a bone defect that was created in the mandible of a rabbit model.

Other signaling molecules with promising clinical applications include bone morphogenetic proteins BMP-6 and BMP-7 as well as vascular endothelial growth factor (VEGF). The latter is being studied in regenerative medicine as an angiogenic and tissue growth factor [93,94]. Cells combined with scaffolds can potentially differentiate into specific organs and tissues that allow the formation and regeneration of signaling molecules as described above.

In recent years, iPS cells have drawn much attention in regenerative medicine [2,95]. iPS cells are created by introducing several types of genes known as Yamanaka factors into somatic cells, such as skin cells, that can be easily collected. These pluripotent cells have the ability to differentiate into any cell in the body and hold great promise in regenerative medicine as well as the treatment of intractable diseases. iPS cells have also been examined for their use in cell-seeded scaffolds for bone regeneration [96,97].

Because cells that are seeded onto scaffolds for bone regeneration do not require pluripotency as found in iPS cells, studies have been conducted using BMSC [98,99]. In clinical practice, an ample amount of BMSC can be obtained from the iliac bone marrow with relative ease. BMSC are suitable for bone regeneration and can be differentiated into osteoblast progenitor cells by culturing in an osteogenic medium containing β-glycerophosphate or dexamethasone [100]. Studies have been conducted for culturing BMSCs onto scaffolds in order for cells to differentiate into osteoblast progenitors and promote bone formation. Liu et al. [67] reported on a chitosan/HA composite, and Xu et al. [101] reported on a PCL/PLLA composite. Both studies described the use of scaffolds combined with BMSC for bone regeneration.

The combined use of scaffolds and platelet-rich plasma (PRP) is also being investigated. PRP is rich in growth factors such as VEGF, insulin-like growth factor (IGF), platelet derived growth factor (PDGF), and transforming growth factor beta (TGF-β). In the context of regenerative medicine, PRP is not used as cells to differentiate into tissues, but as a DDS for signaling molecules [102,103]. Cheng et al. [104] repaired a critical-size cranial defect in a rat model using an implant comprised of a silk fibroin/PCL composite scaffold with PRP.

## 4. Structure of Scaffolds

There has been substantial research on various scaffolds for bone regeneration. Beyond the material of the scaffold itself, an important factor for consideration is its structural characteristics. We have previously combined rhBMP-2 with a hydrogel-like PLA-DX-PEG implant to form ectopic bone in the dorsum of mice [79] and to treat an ulnar bone defect in a rabbit model. The gelatin scaffold reported by Hokugo et al. [57] that incorporated PRP was also a hydrogel. Re et al. [105] cultured BMSC in gelatin-chitosan hybrid hydrogels and reported good cell proliferation as well as differentiation into osteoblasts.

There have been numerous reports on freeze-drying hydrogels in order to form spongy scaffolds. Takeda et al. [106] used the freeze-drying method to fabricate a collagen/rhBMP-2 composite to create a sponge-like implant for reconstructing rat columellae. Takahashi et al. [107] cultured rat mesenchymal stem cells (MSC) on a gelatin/β-TCP composite sponge and observed using SEM that the MSC had penetrated and adhered to the pores of sponge. They confirmed that the ALP activity was increased by using the *p*-nitrophenylphosphate method [108] and that the OCN content was increased by using enzyme-linked immunosorbent assay (ELISA).

In addition to sponge-like structures, scaffolds can also be fabricated with fibrous structures. Nanoscale fibers can be produced with the electrospinning method, whereby a high-voltage electric force is used to draw charged threads of polymer solutions from a nozzle [109]. Lee et al. [110] produced a collagen/PCL composite fiber with an approximate diameter of 350 nm. The nanofiber scaffold, which provides excellent cell adhesion and proliferative capacity, was solidified with PCL to enhance the mechanical strength of the scaffold. MG63 cells were cultured onto this scaffold and showed better cell growth than the PCL scaffold in the MTT assay. The aforementioned collagen/PLLA composite scaffold reported by Zhang et al. [85] and the PVA/HA composite scaffold reported by Enayati et al. [88] were also nanofibrous scaffolds produced by electrospinning

There are several reports regarding the size and shape of the pores in scaffolds. Roosa et al. created PCL scaffolds with different pore sizes (350, 550, and 800 μm) and conducted an experiment to form ectopic bone under dorsal muscle fascia of mice using BMP-7 as signaling molecules. In in vivo studies, porosity did not have a significant effect [111]. In contrast, Kook et al. produced scaffolds with 3D fibrous PCL with a diameter of approximately 300 μm. MC3T3-E1 cells were cultured on the scaffold with gaps between the fibers that ranged between 150 and 350 μm. In their experiments, cell viability and ALP activity were significantly higher when the pore size was narrower [112]. The porosity of the scaffold may be a factor that controls bone delivery, adhesion, and nutrition that may affect bone formation.

There have also been efforts to use spherical particles as scaffolds. Wang et al. evaluated the bone forming capability of porous nanohydroxyapatite-based collagen scaffolds by including insulin-loaded PLGA particles in its pores. The evaluation was performed using three types of particles: nanospheres of 121.62 ± 2.5 nm, microspheres of 1.61 ± 0.08 μm, and 10× microspheres of 21.45 ± 0.22 μm. As a result, ALP and OCN expression from BMSCs was the highest in the microspheres group, indicating a strong differentiation into bone tissue [113]. In an in vivo study, the repair of critical size bone defects were evaluated in rabbit mandibles, and the greatest bone formation was similarly found in the microspheres group. The authors also reported on insulin-loaded PLGA microspheres that were injected into rabbit mandibles around a titanium implant, resulting in good bone formation [114]. We think that bone formation was promoted by the effect of the gradual local release of insulin and easier cell adhesion to suitably-sized spherical particles.

In recent years, scaffolds have been created by 3D printing that enables greater control of their microstructures. The aforementioned PLGA/HA/chitosan scaffold by Deng et al. was created by 3D printing, and its pore size was 431.31 ± 18.40 μm [92]. Zhang et al. [115] developed a PTG implant using 3D printing technology, which was composed of a PLGA and βTCP composite with graphene oxide (GO). The implant featured a latticed structure and pore size of 400 ± 50 μm. Rat MSCs were cultured onto this scaffold, and the gene expression of ALP, OCN, and osteopontin (OPN) were increased, all of which are bone formation markers. The authors were able to repair a rat cranial bone defect with the scaffold.

Factors that affect the performance of biodegradable polymers include both the ease of release and the release rate of signaling molecules. For the optimal performance of scaffolds, it is necessary for signaling molecules to stay within the scaffold for a certain amount of time and to be released at an appropriate speed. Yan et al. [70] described the release of rhBMP-2 from a hyaluronic acid hydrogel-based scaffold, and found that the rhBMP-2 release is regulated by electrostatic and van der Waals forces that can be adjusted by manipulating the pH level. Kim et al. [116] produced a hydrogel composed of (methoxy) PEG-PCL-PLA (MP) copolymer and evaluated the performance of BMP-2 as a carrier. A positively charged MP copolymer derivatized with an amine group (MP-NH2) was prepared for an electrically neutral MP copolymer, and anionic BMP-2 was combined with each of the MP copolymers for comparison. The release of BMP-2 from MP-NH2 was slower than that of MP copolymers. This was probably due to the binding of cationic MP-NH2 and anionic BMP-2 that became stronger in addition to the suppression of release. In in vitro tests of human turbinate mesenchymal stem cells (hTMSCs) cultured on scaffolds, BMP-2/MP-NH2 had a higher expression of the bone formation marker OCN, type I collagen, and OPN than BMP-2/MP copolymer scaffold. Similar results were obtained in in vivo tests that compared BMP-2/MP copolymer and BMP-2/MP-NH2 injected into the dorsum of mice. It was suggested that the controlled release rate of signaling molecules plays an important role in the performance of scaffolds.

Scaffolds for bone tissues should have the following characteristics: an optimal release rate of signaling molecules, good cell adhesion to allow efficient cell seeding and localized induction, and the ability to rapidly get replaced by bone tissue. Various aspects of scaffold structure have been studied and considered, including suitable materials for bone tissue regeneration, pore size, and three-dimensional structure [117,118]. However, there is still no broad consensus on their ideal traits, including the optimal material for bone tissue regeneration, sustained release and biodegradability, and use of a three-dimensional structure. If these structural features for bone tissue regeneration are clarified, it may be possible to create scaffolds that exceed the calcium phosphate-based artificial bones that are widely used in clinical practice today.

## 5. Conclusions

Biodegradable scaffolds are being investigated as DDS in regenerative medicine; in addition, the materials have also been studied for bone tissue regeneration. Although some of the materials perform well at an experimental level, the optimal conditions for bone regeneration have yet to be determined.

In general, the osteoconductive capability of biodegradable polymers is lower than that of commercially and clinically-available calcium phosphate-based artificial bones, and the repair of large bone defects has not been achieved with polymers alone. Some biodegradable polymers have excellent characteristics including their production methods and biocompatibility, and new clinical applications are expected to emerge in regenerative medicine. Due to the large variety of biodegradable polymers, there are innumerable factors to consider for their potential and suitable use as scaffold material. These factors include, but are not limited to, the combination and structure of various bio-based and synthetic polymers, the types of signaling molecules to combine with the material, and the cells to be embedded in the material. A clarification of these conditions could lead to the development of scaffolds for bone regeneration with high efficiency.

## Figures and Tables

**Figure 1 pharmaceutics-12-00095-f001:**
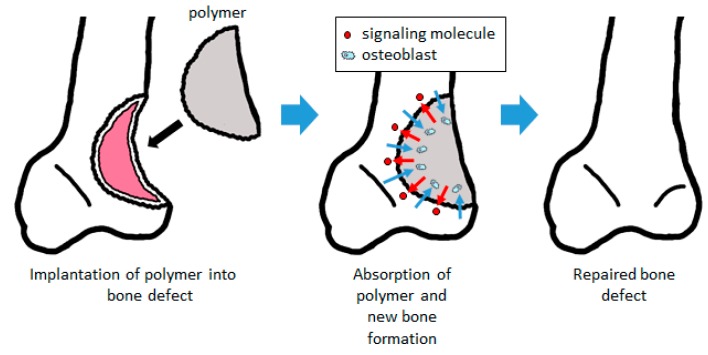
Bone regeneration using a biodegradable polymer scaffold. By filling the bone defect, the signaling molecules contained in the scaffold are released to promote bone formation. Cells such as osteoblast are induced into the scaffold, forming bone tissue and absorbing the scaffold. The scaffold is subsequently replaced with autologous bone.

**Figure 2 pharmaceutics-12-00095-f002:**
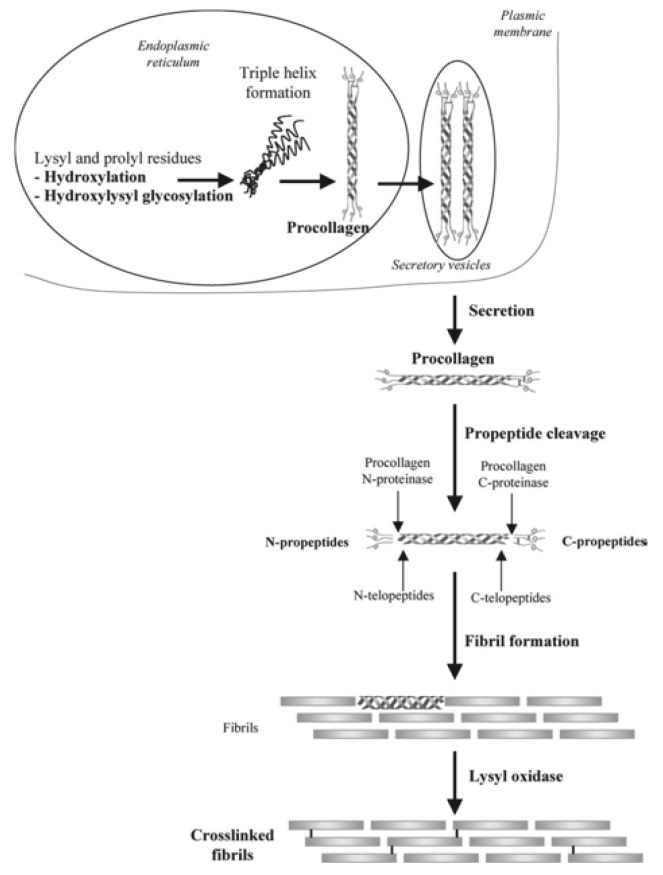
Formation and structure of type I collagen. The N-terminus and C-terminus of procollagen (the precursor of collagen formed in the cell) are secreted out of the cell and subsequently cleaved by protease to become a collagen molecule. Collagen molecules associate to form collagen fibrils, which are reinforced by cross-linking between the collagen molecules to become collagen. Images are modified from a study by Viguet-Carrin et al. [43]. Reproduced with permission from Springer Nature, 2006.

**Figure 3 pharmaceutics-12-00095-f003:**
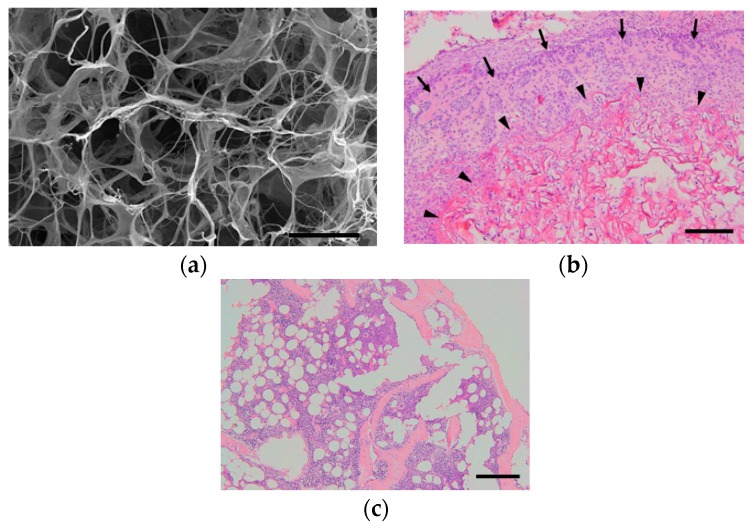
Our experiment on the ectopic bone formation in the dorsum of mice with collagen-based scaffolds. (**a**) Scanning electron microscope image of a freeze-dried collagen scaffold. Scale bar, 50 μm. (**b**) Histopathological image at one week after surgery. The collagen is combined with recombinant human bone morphogenetic protein-2 (rhBMP-2) and transplanted onto the dorsal muscle fascia of mice. Collagen fibers (arrow heads) remained, while woven bones (arrows) were formed around the fibers. Scale bar, 200 μm. (**c**) At three weeks after surgery, no collagen fibers were observed, and mature bones consisting of cortical bone, cancellous bone, and bone marrow tissue were formed. Scale bar, 200 μm.

**Figure 4 pharmaceutics-12-00095-f004:**
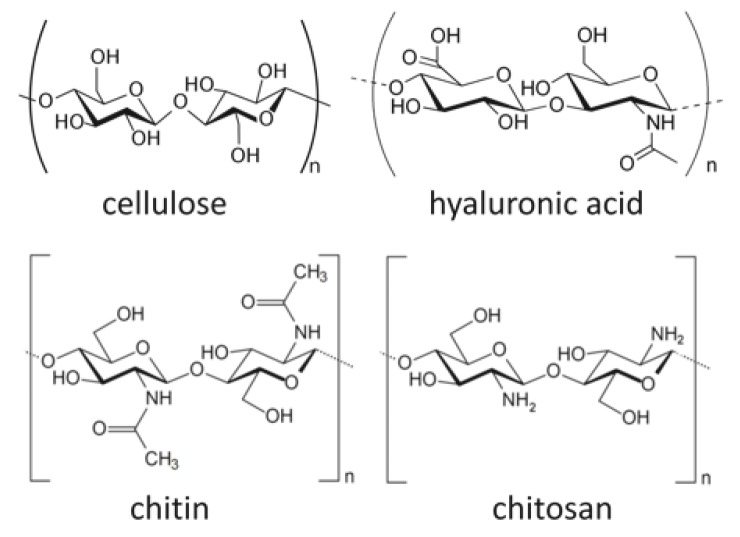
A schematic diagram of some bio-based polymers: cellulose, hyaluronic acid, chitin, and chitosan. Images are modified from a study by Younes et al. [64].

**Figure 5 pharmaceutics-12-00095-f005:**
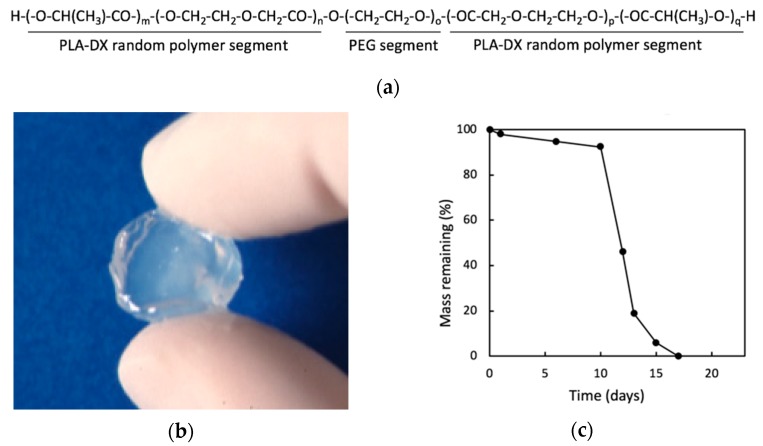
(**a**) Structural formula of poly lactic acid-*p*-dioxanone-polyethylene glycol block copolymer (PLA-DX-PEG). (**b**) Macroscopic image of PLA-DX-PEG hydrogel. (**c**) In vitro solubility curve of PLA-DX-PEG polymer. A 500 mg sample was immersed in PBS at 37 °C, and its weight was measured over time. The weight decreased rapidly after 10 days and was completely dissolved within 20 days. Images are modified from a study by Saito et al. [79]. Reproduced with permission from Springer Nature, 2001.

**Figure 6 pharmaceutics-12-00095-f006:**
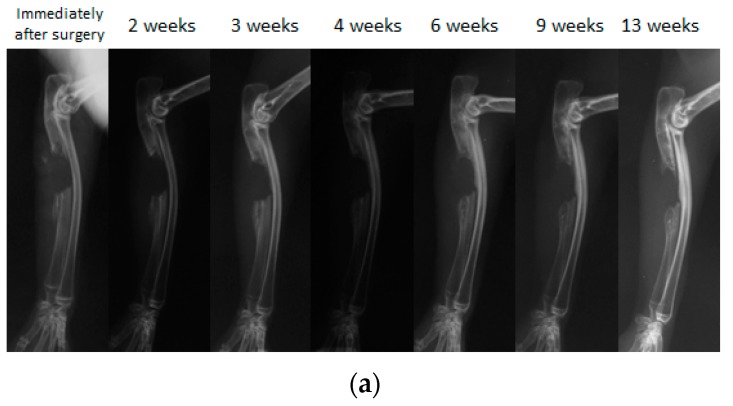
A simple radiographic image of a rabbit ulnar bone defect model. (**a**) A bone defect was created and nothing was filled in the defect. Even 13 weeks after surgery, the defect was not repaired despite the formation of small new bone that was observed in the bone stump. (**b**) Only the PLA-DX-PEG implant was filled in the bone defect. At three weeks after surgery, formation of new bone from the bone stump was observed. At six weeks after surgery, the new bone from the distal stump showed bony union with the rib. Even 13 weeks after surgery, the bone defect resulted in a non-union with no cross-linking in the new bone. (**c**) The bone defect was filled with a PLA-DX-PEG implant combined with 10 μg rhBMP-2. Although formation of new bone from the bone stump was observed, the bone defect of the ulna resulted in a nonunion at 13 weeks after surgery without any cross-linking in the new bone. (**d**) The bone defect was filled with a PLA-DX-PEG implant combined with 100 μg of rhBMP-2. Three weeks after surgery, the bone defect was cross-linked by the newly formed bone from the bilateral bone stumps. Nine weeks after surgery, there was formation of new bone in the bone defect that developed into mature bone with trabecular structure.

**Figure 7 pharmaceutics-12-00095-f007:**
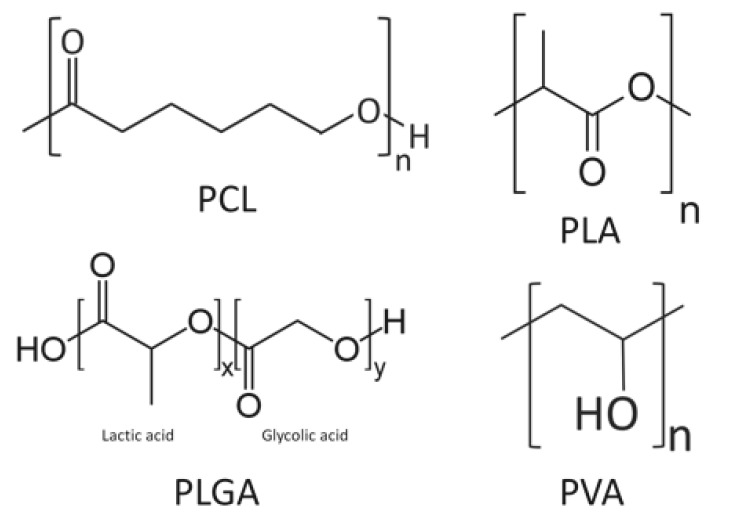
A schematic diagram of some synthetic polymers: polycaprolactone (PCL), poly-l-lactic acid (PLA), polylactic-*co*-glycolic acid (PLGA), and poly (vinyl alcohol) (PVA).

**Table 1 pharmaceutics-12-00095-t001:** Polymers used to produce scaffolds for bone tissue engineering.

Natural polymer	collagen, gelatin, cellulose, chitosan, hyaluronic acid
Synthetic polymer	PCL, PEG, PLLA, PLGA, PVA

PCL: polycaprolactone, PEG: polyethylene glycol, PLLA: poly-l-lactic acid, PLGA: polylactic-co-glycolic acid, PVA: poly (vinyl alcohol).

**Table 2 pharmaceutics-12-00095-t002:** A summary of the most relevant literature on bio-based polymers.

Author, Year	Polymer	Composite	Structure	Cells/Animals	Signaling Molecules	Evaluations
Carstens et al., 2005 [48]	collagen	-	sponge	porcine	rhBMP-2	mandibular bone defect
Yeo et al., 2011 [51]	collagen	βTCP, PCL	nanofiber	MG-63 osteoblast-like cell	-	MTT assay
Sotome et al., 2016 [16]	collagen	HA	sponge	human	-	marketed product
Yokota et al., 2001 [56]	gelatin	PLGA	sponge	Fischer rat	rhBMP-2	ectopic bone
Hokugo et al., 2005 [57]	gelatin	-	hydrogel	rabbit	PRP	ulnar bone defect
Rohanizadeh et al., 2008 [58]	gelatin	-	sponge	MG-63 osteoblast-like cell	-	ALP activity, entry of cells
Chakraborty et al., 2019 [62]	cellulose	-	nanofiber	MC3T3-E1 osteoblast cell	-	MTT assay, SEM
Sharifi et al., 2018 [(66]	chitosan	PCL	nanofiber	MG-63 osteoblast-like cell	-	MTT assay
Liu et al., 2013 [67]	chitosan	HA	nanofiber	BMSC SD rat	BMSC	ALP stain cranial bone defect
Yan et al., 2018 [70]	hyaluronic acid	-	hydrogel	SD rat	rhBMP-2	ectopic bone
Paidikondala et al., 2019 [71]	hyaluronic acid	hydrazone	hydrogel	SD rat	rhBMP-2	ectopic bone

rhBMP-2: recombinant human bone morphogenetic protein, TCP: tricalcium phosphate, PCL: polycaprolactone, MTT assay: 3-(4,5-dimethylthiazol-2-yl)-2,5-diphenyl tetrazolium bromide assay, HA: hydroxyapatite, PLGA: polylactic-co-glycolic acid, PRP: platelet-rich plasma, ALP: alkaline phosphatase, SEM: scanning electron microscopy, BMSC: bone marrow-derived mesenchymal stem cell.

**Table 3 pharmaceutics-12-00095-t003:** A summary of the most relevant literature on synthetic polymers.

Author, Year	Polymer	Composite	Structure	Cells/Animals	Signaling Molecules	Evaluations
Saito et al., 2001 [79]	PLA-DX-PEG	-	hydrogel	ddY mouse	rhBMP-2	ectopic bone
Present study	PLA-DX-PEG	-	hydrogel	JW rabbit	rhBMP-2	ulnar bone defect
Wang et al., 2018 [82]	PCL	nanosilicate	nanofiber	MC3T3-E1 cell SD rat	MC3T3-E1 cell	ALP activity, OCN expression
Yang et al., 2018 [84]	PLGA	nanosilicate	nanofiber	SaOS-2 cell	-	Alizarin Red S stain, ALP activity
Zhang et al., 2013 [85]	PLLA	collagen	nanofiber	MC3T3-E1 osteoblast cell rabbit	-	OCN gene expression
Enayati et al., 2018 [88]	PVA	HA	hydrogel	MG63 cell	-	MTT assay, Alizarin Red S stain

PLA-DX-PEG: poly lactic acid-*p*-dioxanone-polyethylene glycol block copolymer, rhBMP: recombinant human bone morphogenetic protein, PCL: polycaprolactone; ALP: alkaline phosphatase, OCN: osteocalcin, PLGA: polylactic-co-glycolic acid, PLLA: poly-l-lactic acid, PVA: poly(vinyl alcohol). HA: hydroxyapatite, MTT assay: 3-(4,5-dimethylthiazol-2-yl)-2,5-diphenyl tetrazolium bromide assay.

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
