# Peer review of "Biodegradable Polymers as Drug Delivery Systems for Bone Regeneration"

_pharmaceutics, 2020, doi:10.3390/pharmaceutics12020095_

Round 1
Reviewer 1 Report
Table 1: acronyms for synthetic polymers are used before they are defined in the text.
Line 59: in my opinion, it should be "each with its own advantages ...". I suggest to check.
Line 63: it should be "that functions", since the subject is "sufficient material".
Figure 2: in the caption, poly-L-lactic acid should be abbreviated as "PLLA", since PLA can be also intended as the polymer from racemic mixture.
This also allows to differentiate from poly-D-lactic acid (PDLA).
Line 137: no acronym is given for poly(D,L-lactic-co-glycolic acid). I assume it is PLGA, which was used in Table 1 without previous definition.
Line 159: chitosan can be found with a different degree of deactylation, but this is not discussed in the manuscript.
Figure 7: PLA acronym is defined for poly-L-lactic acid, while in line 260 it is used for a generic polylactic acid. They are potentially not the same material
(see my comment above). The authors should harmonize the employed notation. This may also reflect that the authors are not fully aware of materials peculiarities,
which should not happen since they are writing a review.
Table 2 in page 12 should be Table 3. Check the numbering. In addition, "Current study" should be changed, since it is not informative.
Reviewer 2 Report
Good enough for publication
Author Response
Pleese see the attachment.

Reviewer 3 Report
The manuscript is now good written and deserve to be published. Some minor issues are mentioned below. Anyway, one modification is mandatory.
I suggest modifying the table 1 caption because now is too general (Table 1. Biodegradable polymer scaffolds. )- suggestion “Polymers used to made scaffolds for bone tissue engineering”
Avoid “etc” used in table 1.
I suggest modifying the figure legend for figure 1 who is wrong (“A schema of an ideal biodegradable polymer scaffold.” is not represented in this figure.
I suggest modifying the figure legend for figure 2 who is wrong (“Schematic diagram of synthetic polymers.” is partially represented in this figure. I suggest to use “…some synthetic polymers…”. Similar issue with figure 4 and figure 7.
I suggest modifying the table 2 caption because now is too general (Table 2. Literature on bio-based polymers.)
Is mandatory to modify the number of table from 296 line (Table 3 instead of Table 2). Also, the same problem with the table caption who is too general.
Reviewer 4 Report
Dear Editor, Dear Authors,
In this manuscript Aoki and Saito review the literature on the use of biodegradable polymers as scaffolding materials for bone regeneration and as drug delivery systems for signalling molecules that can induce bone regeneration.
The review is well written, concise and objective. The experience and expertise of the authors in this field, results in a critical selection and analysis of the literature that will be of great benefit for readers interested in regenerative medicine, in general, and bone regeneration, in particular.
In the new version of the manuscript the authors addressed suitably my former comments. Summarising the literature covered in each section in a table format is also very helpful for readers.
I believe that the new version is ready for publication after correction of the caption of Figure 2, which is in fact the caption of Figure 7.
Best regards
Author Response
Please see the attachment.

This manuscript is a resubmission of an earlier submission. The following is a list of the peer review reports and author responses from that submission.
Round 1
Reviewer 1 Report
This was a relatively well written review of polymers for bone regeneration in terms of the organization and grammar; however, many of the topics discussed felt somewhat cursory without much depth of analysis. Additionally, many of the references were more than 3 years old. There were also a couple of sentences that were poorly worded (as outlined in the attached manuscript).

Reviewer 2 Report
The authors wrote a review aimed at illustrating the use of both synthetic and natural polymers as raw materials for scaffolds for bone regeneration.
The contents of the review are generally appropriate (the Introduction paragraph is well written) and I am aware that this paper can only "scratch the surface", given the extent of the topic.
Anyway, I feel that the discussion can be still improved and expanded. Indeed, there is a similar review on the same journal (Pharmaceutics) that offers a broader overview (Ferracini et al., Scaffolds as structural tools for bone-targeted drug delivery, Volume 10, Issue 3, 2018). I would like to highlight some points that require attention:
In table 1, PLGA should be defined as polylactic-co-glycolic acid, since it is a copolymer and not a blend. The definition provided by the authors can be misleading. The title of the review is "Biodegradable Polymers as Drug Delivery Systems for Bone Regeneration" but materials themselves are poorly discussed. In my opinion, the authors should provide a better description of the materials and why they are used in this field. While natural polymers are somehow discussed, a description of the main peculiarities of synthetic polymers is completely absent. The authors do not mention, e.g., the intrinsic biocompatibility of aliphatic polyesters and their in situ degradation through hydrolysis, which are some of the main factors that determined their success. The authors should insert new figures with the chemical structures of the discussed polymers. The authors did not consider hyaluronic acid as raw material. There are some interesting examples, like this ones: Yan et al., Synthetic design of growth factor sequestering extracellular matrix mimetic hydrogel for promoting in vivo bone formation, Biomaterials, 2018, 161, 190 - 202; Paidikondala et al., Impact of Hydrogel Cross-Linking Chemistry on the in Vitro and in Vivo Bioactivity of Recombinant Human Bone Morphogenetic Protein-2, ACS Applied Biomaterials, 2019, Volume 2, Issue 5, 2006 - 2012. Concerning drug delivery, it would be nice to know if the optimal release profile is known and how to change the release rate. For example, in the Yan et al. paper previously mentioned it is shown that electrostatic interactions between BMP2 protein and the gel can change the release rate and that a slower rate improves bone formation. This was shown also in this paper: Kim et al., An injectable cationic hydrogel electrostatically interacted with BMP2 to enhance in vivo osteogenic differentiation of human turbinate mesenchymal stem cells, Materials Science and Engineering C, 103, 2019. At the end of each paragraph, it would be useful to put tables that summarize all the discussed examples.Reviewer 3 Report
In this manuscript Aoki and Saito review the literature on the use of biodegradable polymers as scaffolding materials for bone regeneration and as drug delivery systems for signaling molecules that can induce bone regeneration.
The review is well written, concise and objective. The experience and expertise of the authors in this field, results in a critical selection and analysis of the literature that will be of great benefit for readers interested in regenerative medicine, in general, and bone regeneration, in particular.
In my opinion the manuscript is clearly worth publication after addressing some minor points:
The authors should include a list of the abbreviations that are used throughout the manuscript. For example, in the caption of Figure 2 the acronym rhBMP-2 is used but not defined before. It would be very helpful for readers to have access to the chemical structure of the natural and synthetic polymers mentioned in table I and throughout the manuscript. The authors could include in the manuscript a chart with the chemical structures. Figure 2 is not referenced neither in the text nor in the figure caption. Is this original work from the authors or taken from a published work? In line 105 the authors describe the use of “type I collagen” as scaffold for bone regeneration. Above, the authors describe the use of collagen as a bone scaffold. The authors should include in the beginning of the section a short introduction on t the structure of collagen. In line 138 the authors state that “chitosan is found in exoskeleton of crustaceans …” . This is imprecise. In fact chitin is the polyssaccharide found in exoskeleton of crustaceans, insects etc. Chitosan is obtained by chemical/enzymatic deachetylation of chitin. Line 153. There seems to be some repetition: Synthetic polymers are gaining popularity as scaffold material for bone regeneration. And “Synthetic polymers are also attracting attention as scaffolds for bone regeneration.” Figure 4 needs to be referenced to in the text and in the caption. Is this original work from the authors or taken from a published work? The images in Figure 4c show a different disposition in comparasion to images 4a, 4b and 4d. why? This needs clarification. In subsection “2.3 composite Polymers” the authors describe the use of composite polymers, but they have described above in subsection 2.2 many composite systems. To me this separation is redundant. Some phrases seem to me “cumbersome” and difficult to understand- maybe just a question of writing stile…Reviewer 4 Report
The author provides a comprehensive review. However, there are still some issues needed revision.
The author just listed various polymers and their applications. We all know the advantages, but their disadvantages are lacked. If the cited publications do not mention it, the author should discuss it in his opinion. More schematic diagrams or figures are needed to demonstrate the cited results more visually. Another interesting research section that the author should add in the manuscript to is the form of polymers. The paper mainly reviews the development of polymers in the form of scaffold. However, microsphere is another one that is widely studied. Referring following publications may give the author some information: 1) The effect of a single injection of uniform-sized insulin-loaded PLGA microspheres on peri-implant bone formation, X Wang, L Wang, F Qi, J Zhao, RSC advances 8 (70), 40417-40425. 2) Porous nanohydroxyapatite/collagen scaffolds loading insulin PLGA particles for restoration of critical size bone defect, X Wang, X Wu, H Xing, G Zhang, Q Shi, L E, N Liu, T Yang, D Wang, F Qi, ACS applied materials & interfaces 9 (13), 11380-113. Are there any products available products in the market? If yes, the products should be discussed in the manuscript. If no, what is the reason. The author should have more discussion and his own opinions on the topic rather than just citing papers, no matter it is right or wrong.Reviewer 5 Report
The manuscript "Biodegradable Polymers as Drug Delivery Systems
for Bone Regeneration" is not good to be published due to many missing points. There are many synthetic polymers (like polycaprolactone) and polymer based composites that are not mentioned by the authors. Also, I not found a schematic presentation (eventually a schematic figure) or classifications of these materials or of the drugs intended to be delivered.
Also, regarding the structure of the scaffolds, I expect to find more aspects and correlations with processing technologies and their influence on the structure. On the other hand, the problems of porosity is very important when we discuss about drug delivery systems for bone regeneration.
The methods used for scaffolds characterization is not described or mentioned.
In conclusion, I think that the manuscript is not suitable to be published.